# Narrow-spectrum resource-utilizing bacteria drive the stability of synthetic communities through enhancing metabolic interactions

Wei Wang, Yanwei Xia, Panpan Zhang, Mengqing Zhu, Shiyi Huang, Xinli Sun, Zhihui Xu ®, Nan Zhang ®, Weibing Xun ®, Qirong Shen ®, Youzhi Miao ® ✉ & Ruifu Zhang ®

The importance of synthetic microbial communities in agriculture is increasingly recognized, yet methods for constructing targeted communities using existing microbial resources remain limited. Here, six plant-beneficial bacterial strains with distinct functions and rhizosphere resource utilization profiles are selected to construct stable, multifunctional communities for plant growth promotion. Metabolic modeling reveals that narrower resource utilization correlates with increased metabolic interaction potential and reduced metabolic resource overlap, contributing to greater community stability. Integrated analyses further consistently confirm the central roles of narrow-spectrum resource-utilizing strains, *Cellulosimicrobium cellulans* E and *Pseudomonas stutzeri* G, which form metabolic interaction networks via secretion of asparagine, vitamin B12, isoleucine, and their precursors or derivatives. Two synthetic communities, SynCom4 and SynCom5, have high stability in the tomato rhizosphere and increase plant dry weight by over 80%. Our study elucidates the relationship between resource utilization width and community stability, providing a rational strategy for designing stable, multifunctional microbial communities for specific habitats.

Microorganisms play essential roles in elemental cycling[1], nutrient activation[2], disease suppression[3], and plant growth promotion[4]. In natural ecosystems, microbes form complex communities with superior productivity, resource efficiency, metabolic complexity and resilience against external disturbances[5–7]. Research on these microbial interactions, particularly coexistence and competition, has underscored their importance in sustaining community functionality and stability[8,9]. While such research has greatly advanced our understanding of agricultural production[10] and demonstrated the potential applications of microbial resources, the functional instability of applied individual strains remains unresolved. This is mainly due to our incomplete understanding of the intricate dynamics and interaction mechanisms within natural microbial communities. Over the past two

decades, advances in sequencing technologies have led to the discovery of a vast number of microorganisms, with 16S rDNA and ITS sequencing unveiling the species diversity and function in different natural environments[11–14]. With over 350,000 bacterial and 40,000 fungal genome sequences now abailable[15], alongside breakthroughs in microbial culture techniques[16], a more integrated systems biology approach has emerged. This has driven a shift from single-strain applications towards synthetic microbial communities engineered to exhibit enhanced stability and functional efficiency.

Current strategies for constructing synthetic microbial communities are categorized as either top-down or bottom-up[17,18]. The top-down approach is based on the principle of simplifying natural communities and optimizing them under specific habitats, with the aim of

Jiangsu Provincial Key Lab of Solid Organic Waste Utilization, Key lab of organic-based fertilizers of China, Jiangsu Collaborative Innovation Center of Solid Organic Wastes, Educational Ministry Engineering Center of Resource-saving fertilizers, Nanjing Agricultural University, Nanjing, China. ✉ e-mail: yzmiao@njau.edu.cn

enhancing outcomes such as crop production[19] and pathogen suppression[20]. However, this approach does not allow for precise control over species composition and functional diversity and is, therefore, limited in its application in complex, large-scale agricultural systems. In contrast, the bottom-up approach was originally designed to enable the customized assembly of microbial communities. This method requires consideration of various factors, including, but not limited, to environmental conditions (nutrients, temperature, and humidity)[21,22], antagonistic and competitive interactions[23,24], cross-feeding relationships[25,26], phylogenetic diversity, and species abundance. By fostering cooperation and minimizing competition, the bottom-up approach enables the targeted construction of stable and multifunctional communities. Nonetheless, there are still some challenges with this approach due to the complexity of microbial interactions, and theoretical frameworks for such bottom-up designs are still under development, with no universally effective operational methods currently in place.

The core principle for constructing stable synthetic communities is to enhance cooperation between microbial members and reduce internal competition[27]. Central to this strategy is optimizing the relationship of external resource utilization among members. However, Due to differences in genome size, microorganisms exhibit significant variations in their capacity to utilize external resources, as well as in the diversity of resources available to them[28]. These differences, inevitably, have complex effects on the stability of synthetic communities. Genome-scale metabolic models (GMMs) provide a powerful tool for studying microbial communities[29], and recent advances in GMM and high-throughput analysis have revealed key ecological mechanisms within microbial communities, such as metabolic dependencies, that drive species co-occurrence and the interplay between competition and cooperation in natural communities[30,31]. These models have also demonstrated significant potential in various habitats, including phototrophic communities[32], plant-microbe interactions[33], and the human gut[34]. Thus, GMMs might have potential applications in the optimal design of synthetic microbial communities. Notably, metabolic modeling has identified two critical metrics—metabolic resource overlap (MRO) and metabolic interaction potential (MIP)—which are pivotal for determining community coexistence and stability[30]. Therefore, clarifying the relationships between microbial resource utilization, cooperation and competition dynamics, and community coexistence and stability in specific habitats, while integrating high-throughput metabolic modeling into bottom-up strategies, holds significant potential for optimizing the construction of stable and multifunctional communities with more predictable and efficient outcomes.

In this study, we develop a strategy for constructing stable and multifunctional synthetic rhizosphere microbiome. Strains exhibiting key plant beneficial functions, such as nitrogen fixation, phosphate solubilization, indoleacetic acid (IAA) synthesis, and siderophore

production, are selected as candidates, with significant variation observed in their abilities to utilize common rhizosphere resources. Our results demonstrate that strains with narrow-spectrum resource utilization ability improve community MIP and reduce MRO, thereby favoring metabolic interactions that promote community stability. Based on these findings, we use metabolic modeling for the precise construction of two stable and multifunctional synthetic communities that exhibit high stability in the tomato rhizosphere and achieve over an 80% increase in plant dry weight. We observe a negative relationship between resource utilization width and community stability, thus providing an efficient, GMM-integrated strategy for bottom-up construction of synthetic microbial communities.

## Results

### Candidate plant-beneficial bacteria for constructing synthetic communities

Plant-beneficial bacteria typically exhibit functions such as nitrogen fixation, phosphate solubilization, IAA synthesis, and siderophore production. To construct synthetic communities, we employed a function-driven selection strategy, prioritizing strains with well-established and widely recognized beneficial traits rather than simplifying native microbiomes based on abundance or co-occurrence patterns. Accordingly, six different bacterial strains (*Bacillus megaterium* L, *Pseudomonas fluorescens* J, *Bacillus velezensis* SQR9, *Pseudomonas stutzeri* G, *Cellulosimicrobium cellulans* E, and *Azospirillum brasilense* K)[35–39] isolated from the plant rhizosphere were selected. Among these candidate strains, nitrogen fixation was primarily detected in *A. brasilense* K and *P. stutzeri* G, which exhibited activity levels of 3517 and 890 nmol $C_2H_4$ $h^{-1} \cdot mg^{-1}$, respectively (Fig. 1a). Phosphate solubilization was most prominent in *P. fluorescens* J, which achieved a soluble phosphorus concentration of 46.39 mg·L$^{-1}$ in NBRIP medium, while *P. stutzeri* G, *B. velezensis* SQR9, and *B. megaterium* L displayed moderate phosphate solubilization activities ranging from 25.51 mg·L$^{-1}$ to 30.47 mg·L$^{-1}$ soluble phosphorus in NBRIP medium. *C. cellulans* E and *A. brasilense* K showed negligible phosphorus solubilization activity (Fig. 1b). For IAA synthesis, all strains, except for *C. cellulans* E, exhibited robust IAA production exceeding 40 mg·L$^{-1}$ in low-salt Luria-Bertani medium supplemented with 1 g·L$^{-1}$ tryptophan. *P. stutzeri* G was the most prolific species, producing 66.08 mg·L$^{-1}$ IAA (Fig. 1c). In terms of siderophore production, the *Bacillus* and *Pseudomonas* strains demonstrated higher activities (Fig. 1d). Overall, these bacterial strains possess diverse plant growth-promoting activities, which highlight their value in the assembly of multifunctional plant-beneficial synthetic communities.

### Narrow-spectrum resource-utilizing strains facilitate community cooperation

To construct a stable plant-beneficial multifunctional bacterial community, antagonistic interactions between the candidate strains were

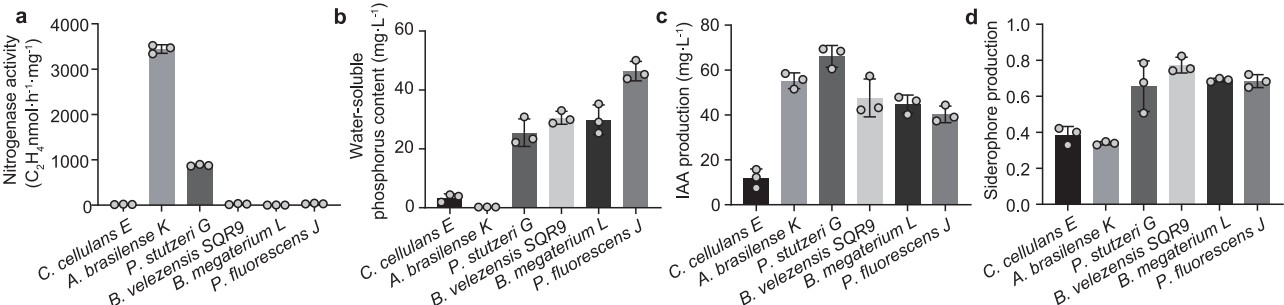

**Fig. 1 | Functional assessment of six plant-beneficial bacteria.** Nitrogenase activity (**a**), phosphorus-solubilizing activity (**b**), indole-3-acetic acid (IAA) production (**c**) and siderophore production (measured as CAS activity) (**d**) of each species were assessed with three biological replicates. Error bars represent mean ± SD of biological replicates (*n* = 3). Source data are provided as a Source Data file.

cross-evaluated and showed negligible effect (Supplementary Fig. 1), indicating that neither interference nor antibiotic competition affected the community stability of the candidate strains. Consequently, our focus shifted to resource-based competition and cooperation through an analysis of their metabolic profiles using phenotype microarrays, targeting the 58 carbon sources commonly existed in the plant rhizosphere. All strains efficiently metabolized fructose, D-glucose, gluconic acid, pyruvic acid, glycerol and lactic acid, while each strain exhibited distinct metabolic capabilities beyond these shared substrates (Fig. 2a). These data were further used to calculate the resource utilization width and overlap, representing carbon substrate diversity and shared resource use, respectively. Results showed that *B. velezensis* SQR9, *B. megaterium* L, and *P. fluorescens* J exhibited the highest resource utilization widths (35.50, 36.76 and 37.32, respectively; Fig. 2b) and average overlap indices (0.83, 0.74, and 0.72, respectively; Fig. 2b), reflecting broad adaptability to rhizosphere resources and substantial similarity in resource use, which may increase their competitive potential toward other species. In contrast, *P. stutzeri* G and *A. brasilense* K had lower resource utilization widths (25.59 and 24.37), while *C. cellulans* E displayed the lowest width (13.10) and overlap index (0.51) (Fig. 2b), indicative of a more specialized metabolic niche, potentially linked to lower competitive pressure and enhanced opportunities for metabolic complementarity. These findings highlight the metabolic heterogeneity among candidate strains and suggest that such metabolic differences may shape their interactions.

To better visualize how differences in resource utilization influence community interactions, we constructed genome-scale metabolic models for each strain, refined using Biolog phenotype data. These models were applied to simulate all 57 possible community combinations (comprising two to six members each), enabling the calculation of two key indices: metabolic interaction potential (MIP), reflecting cooperative potential, and metabolic resource overlap (MRO), indicating competitive pressure. In pairwise communities, narrow-spectrum resource-utilizing (NSR) strains (*C. cellulans* E, *A. brasilense* K, and *P. stutzeri* G) contributed significantly to elevated MIP scores (average 1.53), whereas broad-spectrum resource-utilizing (BSR) strains (*B. velezensis* SQR9, *P. fluorescens* J, and *B. megaterium* L) were associated with lower MIP scores (average 0.6) (Fig. 2c). A clear trend was observed: resource utilization width showed a negative correlation with MIP ($R^2 = 0.4901$, $p < 0.0001$) and a positive correlation with MRO ($R^2 = 0.3465$, $p < 0.001$) (Fig. 2c). Simulated communities with increasing member numbers displayed enhanced cooperation potential (higher MIP and lower MRO scores; Fig. 2d), and NSR strains such as *C. cellulans* E and *P. stutzeri* G consistently acted as central contributors to MIP in multi-member communities (Fig. 2e). Together, these simulations provide a system-level perspective that integrates resource utilization traits and their modeled impact on community-level interactions. This approach highlights a subset of stable, multifunctional communities comprising four to five members, characterized by high MIP and low MRO scores, typically centered around the NSR strains *C. cellulans* E or *P. stutzeri* G.

### NSR strain-driven cooperation is a common pattern in plant-associated bacteria

To assess the generality of NSR strains promoting community cooperation, we further analyzed 224 manually curated metabolic models of *Arabidopsis thaliana* phyllosphere microbiota from Schäfer et al.[40], along with strain-level experimental data on the utilization of 45 phyllosphere carbon sources. Simulations of ~25,000 pairwise interactions revealed that bacterial resource utilization width (simplified as the number of utilizable compounds) was negatively correlated with community MIP scores ($R^2 = 0.0343$, $p < 0.0001$) and positively with MRO scores ($R^2 = 0.0303$, $p < 0.0001$) (Supplementary Fig. 2a). While

directionally consistent with expectations, the low $R^2$ values limited explanatory strength.

Further analysis showed that the experimental resource utilization widths across the 224 strains approximately followed a normal distribution, and genome-based predictions exhibited an even closer fit (Fig. 3a). Experimental and predicted widths were significant positively correlated ($R^2 = 0.4057$, $p < 0.0001$, Fig. 3b), supporting the reliability of genome-informed estimation. We therefore extended the analysis to plant rhizosphere microbiota, incorporating 3001 high-quality bacterial genomes published by Dai et al.[41]. Predicted resource utilization widths in these strains also follow a normal distribution pattern (Fig. 3a). These findings suggest that a stable subset of plant-associated bacteria, beyond the majority of species with moderate resource utilization capacity, tends to adopt ecological strategies involving either markedly reduced or expanded resource use—corresponding to NSR and BSR strains, respectively. Thus, we selected NSR and BSR strains ($\leq 7$ and $\geq 29$ utilizable compounds, respectively) from the tails of the phyllosphere distribution and simulated their pairwise interactions with all 224 phyllosphere strains. In this context, both experimental and predicted resource utilization widths showed significantly negatively correlations with MIP ($R^2 = 0.2749$, $p < 0.0001$; $R^2 = 0.1827$, $p < 0.0001$) and positive correlations with MRO ($R^2 = 0.3915$, $p < 0.0001$; $R^2 = 0.2527$, $p < 0.0001$) (Fig. 3c). Notably, these relationships became stronger with increasing phylogenetic distance among community members (Fig. 3d), consistent with empirical strategies favoring phylogenetic diversity in synthetic community design. In addition, the negative correlation between resource utilization width and MIP was strengthened as community size increased (Supplementary Fig. 2b). To further validate these patterns, we experimentally profiled 25 rhizosphere strains from our laboratory collection for their utilization of 41 carbon sources (Supplementary Data 1). Despite a general bias toward high resource utilization capacity (88% strains $\geq 19$ compounds), experimental resource utilization widths remained significantly negatively correlated with MIP, which also strengthened progressively with increasing community size (Supplementary Fig. 2c).

Building on these findings, we refocused on the six functionally distinct strains selected for this study, which exhibit a gradient in resource utilization width and substantial phylogenetic divergence. The cooperative enhancement by NSR strains such as *C. cellulans* E or *P. stutzeri* G exemplifies the broader ecological trend in plant-associated bacteria.

### NSR strain-driven cooperation sustain the community stability

To assess the contribution of NSR-strain-driven cooperation to community stability, we selected two bacterial communities: SynCom4 comprising four strains of *C. cellulans* E, *B. velezensis* SQR9, *P. fluorescens* J, and *A. brasilense* K, and SynCom5 comprising five strains of *C. cellulans* E, *P. stutzeri* G, *B. velezensis* SQR9, *A. brasilense* K, and *B. megaterium* L. Both communities contained *C. cellulans* E and/or *P. stutzeri* G and have been characterized by the same high MIP scores of 4 and low MRO scores (0.77 and 0.78 for SynCom4 and SynCom5, respectively). Three control communities were also included: SynCom3-1 containing the three strains *B. velezensis* SQR9, *P. fluorescens* J. and *B. megaterium* L; SynCom3-2 comprising the three strains *B. velezensis* SQR9, *P. stutzeri* G, and *B. megaterium* L; and SynCom4-1 containing the four strains *B. velezensis* SQR9, *P. fluorescens* J, *A. brasilense* K, and *B. megaterium* L. None of these control communities contain *C. cellulans* E or *P. stutzeri* G, and they all have low MIP scores (0, 1 and 1, for SynCom3-1, SynCom3-2, and SynCom4-1, respectively) and high MRO scores (0.81, 0.84, and 0.81 for SynCom3-1, SynCom3-2, and SynCom4-1, respectively). In situ inoculation of these SynComs in the tomato rhizosphere showed a rapid decline in the SynCom3-1, SynCom3-2, and SynCom4-1 populations (Fig. 4a). The SynCom3-1 and SynCom3-2 populations

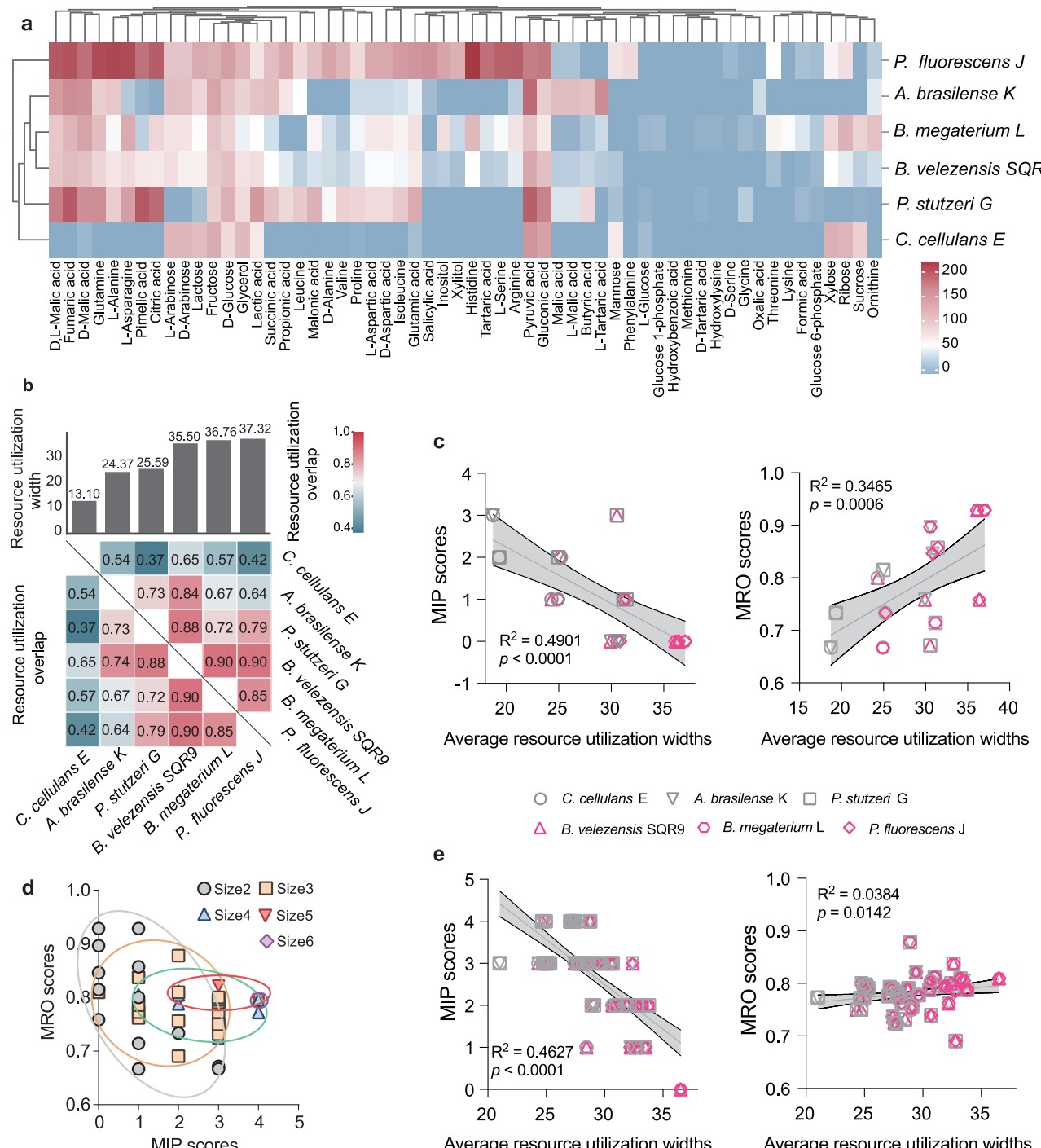

**Fig. 2 | Rhizosphere resource utilization and metabolic interaction simulation of six plant-beneficial strains. a** Heatmap showing the utilization spectrum of 58 common rhizosphere carbon sources by six plant-beneficial strains, with OmniLog values indicating cell respiration intensity. These data were used to calculate each strain's resource utilization width and overlap (**b**). To provide a dynamic and integrative view of the relationship between resource utilization and metabolic interaction, genome-scare metabolic models refined with Biolog data were constructed to simulate all 57 possible community combinations (with two to six members each), enabling the calculation of two key indices: metabolic interaction potential (MIP), indicating cooperative potential, and metabolic resource overlap (MRO), indicating competitive pressure. **c** Correlation between average resource utilization widths and corresponding MIP/MRO scores in the paired communities. **d** Changes in MIP/MRO scores with increasing community size. **e** Correlation between average resource utilization widths and corresponding MIP/MRO scores in communities with three or more members. The line represents a linear regression fit with 95% confidence interval (asymptotic). Source data are provided as a Source Data file.

dropped from $10^7$ cfu·g⁻¹ to below $10^4$ cfu·g⁻¹ within 60 h, with the abundance of *B. megaterium* L in SynCom3-2 dropping sharply to below $10^3$ cfu·g⁻¹. Similarly, the population of all SynCom4-1 members fell below $10^4$ cfu·g⁻¹ within 48 h. In contrast, SynCom4 and SynCom5 exhibit significantly greater stability ($p < 0.01$), with most members maintaining populations above $10^5$ cfu·g⁻¹ at 60 h (Fig. 4b, c). Furthermore, targeted strain dropout from the SynCom communities revealed that removing *C. cellulans* E from SynCom4

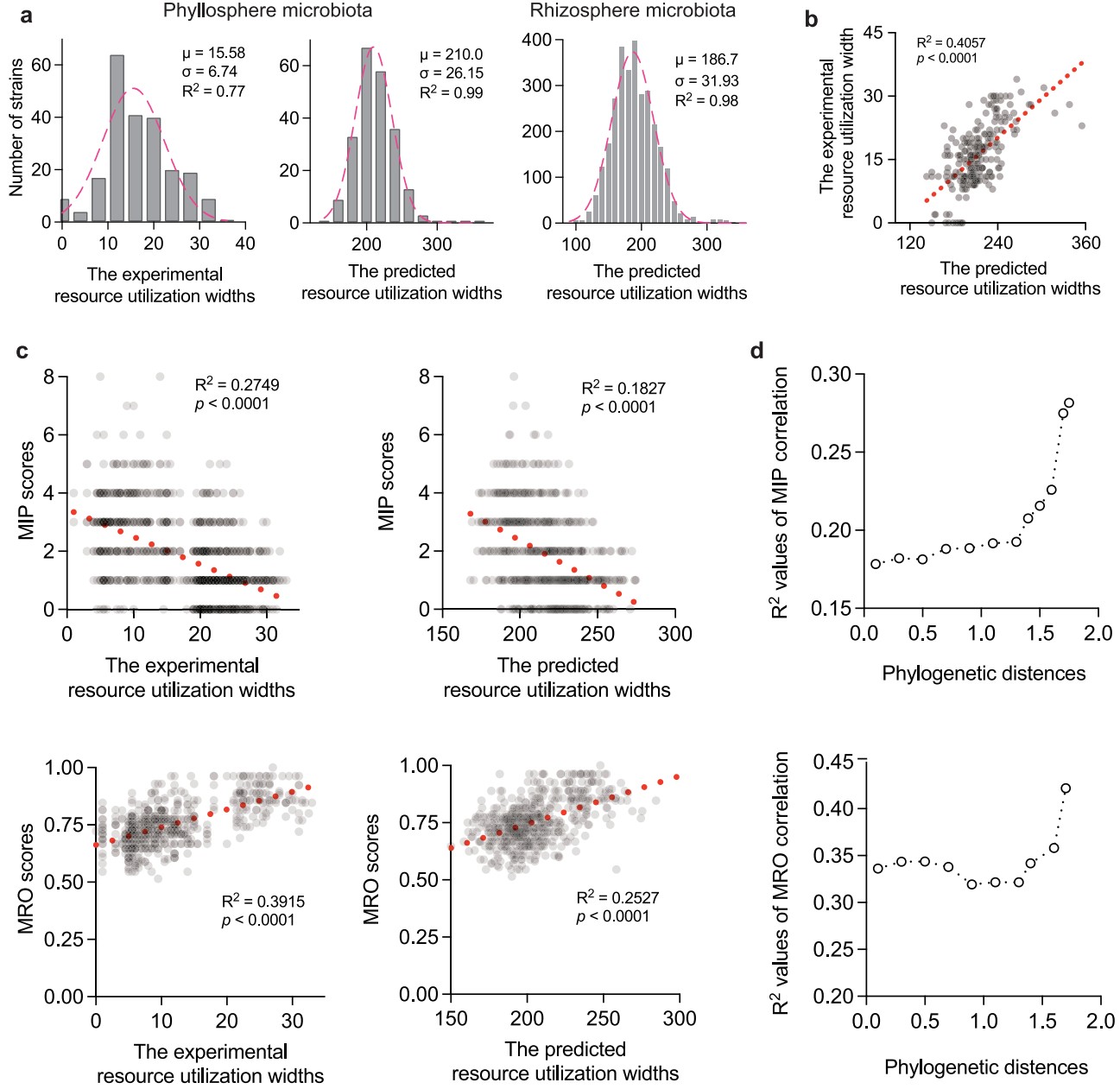

**Fig. 3 | Relationships between resource utilization width and metabolic interaction potential or metabolic resource overlap in plant-associated bacteria. a** Distribution of experimental and genome-predicted resource utilization widths for 224 phyllosphere strains, and predicted widths for 3001 rhizosphere bacterial genomes. Gaussian nonlinear regression curve showing best-fit line. **b** Correlation between experimental and predicted resource utilization widths in phyllosphere strains. **c** Pairwise interaction simulations using narrow-spectrum and broad-spectrum resource-utilizing strains (with resource utilization widths ≤7 and ≥29) against all 224 phyllosphere strains, showing correlations between community metabolic interaction potential (MIP) or metabolic resource overlap (MRO) and average experimental or predicted resource utilization width. The line represents a linear regression fit. **d** Effect of phylogenetic distance on the strength of the correlation ($R^2$) between community MIP/MRO and average experimental resource utilization width. The 224 phyllosphere bacterial models and the 3001 rhizosphere bacterial genomes were derived from the published data of Schäfer et al. and Dai et al., respectively[40,41]. Source data are provided as a Source Data file.

resulted in a decrease in the *A. brasilense* K population from $7.9 \times 10^5$ to $2.0 \times 10^4$ cfu·g⁻¹ (Fig. 4b). Similarly, removing *P. stutzeri* G from SynCom5 resulted in a decrease in the *C. cellulans* E population from $5.0 \times 10^5$ to $1.5 \times 10^4$ cfu·g⁻¹ (Fig. 4c). However, dropout of other members had no significant impact on the stability of the communities in which *C. cellulans* E or *P. stutzeri* G remained (Fig. 4b, c). Overall, we demonstrated that the narrow-spectrum resource-utilizing *C. cellulans* E and *P. stutzeri* G play crucial roles in enhancing MIP and reducing MRO and, thus, make a significant contribution to in situ community stability.

## *C. cellulans* E and *P. stutzeri* G stabilize the community by secreting key metabolites

To elucidate metabolic interactions in the stable SynComs synthesized, cross-feeding assays were conducted using artificial root exudate spent medium (Fig. 5a and Supplementary Fig. 3). The results showed that post-fermentation supernatant from *C. cellulans* E significantly promoted the growth of *B. velezensis* SQR9, *B. megaterium* L, and *A. brasilense* K. Similarly, supernatant from *P. stutzeri* G significantly enhanced the growth of *B. megaterium* L, *A. brasilense* K, and *C. cellulans* E. Further, supernatant from *A. brasilense* K markedly

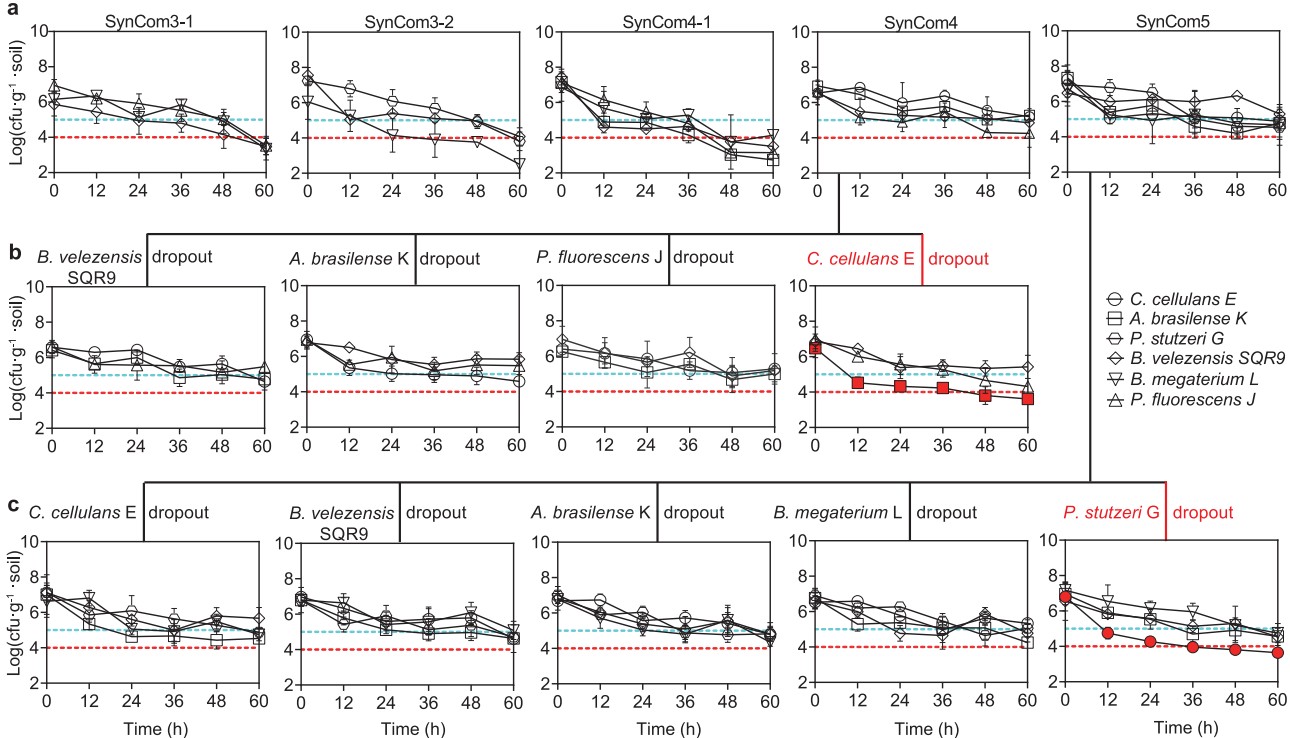

**Fig. 4 | In situ dynamic changes of strain abundances in the synthesized communities.** Synthetic communities with high MIP and low MRO scores (SynCom4 and SynCom5) and control communities with low MIP and high MRO scores (SynCom3-1, SynCom3-2, and SynCom4-1) were inoculated in the tomato rhizosphere. Temporal changes in strain abundance are shown (**a**). Under the same conditions, changes in the remaining strain abundances following individual member dropout are shown for SynCom4 (**b**) and SynCom5 (**c**). The red and blue dashed lines represent strain densities of $1.0 \times 10^4$ and $1.0 \times 10^5$ cfu·g$^{-1}$ soil, respectively. Error bars represent mean ± SD of biological replicates ($n = 3$). Source data are provided as a Source Data file.

promoted the growth of *C. cellulans* E. These results clearly demonstrate that the secretions of *C. cellulans* E, *P. stutzeri* G, and *A. brasilense* K support the growth of other community members. In particular, *C. cellulans* E, which had the narrowest-spectrum of resource-utilization, exhibit extensive cross-feeding. In contrast, the post-fermentation supernatants of *B. velezensis* SQR9, *P. fluorescens* J, and *B. megaterium* L strains, which exhibited broad-spectrum resource utilization, did not support the growth of any other members. These findings confirm the notion that *C. cellulans* E and *P. stutzeri* G serve as metabolic hubs in the stable SynCom4 and SynCom5 communities, thus further supporting the active role of strains with narrow-spectrum resource utilization in the construction of metabolic interaction networks and promotion of community stability.

Given the crucial roles of *C. cellulans* E and *P. stutzeri* G in maintaining the stability of metabolically interconnected communities, we next investigated their contributions to supporting other members. Using high-throughput untargeted metabolomics, we detected over 5600 metabolites spanning nine major chemical classes in the culture supernatants of six individual strains and their respective synthetic communities following 48 h of fermentation on artificial root exudate spent medium (Fig. 5b). Strain-specific cultures displayed distinct extracellular metabolite profiles, with *P. stutzeri* G markedly diverging from the others, suggesting potential for metabolic exchange upon community assembly. Compared to single-strain cultures, SynCom4 and SynCom5 exhibit substantially more diverse and compositionally distinct metabolomes, indicative of extensive exchanges and community-level metabolic reprogramming in multispecies contexts. Dropping out *C. cellulans* E from SynCom4 or *P. stutzeri* G from SynCom5 resulted in profound perturbations of the extracellular metabolome, underscoring their central roles in structuring metabolic exchange networks. Although these results delineate the metabolic

signatures of stable communities centered on *C. cellulans* E or *P. stutzeri* G, the large number of differential metabolites (Supplementary Data 2), including numerous secondary metabolites, complicates the identification of key cross-feeding compounds.

Therefore, we integrated genome-scale metabolic models to calculate the microbial producibility metric (PM) for 88 biomass synthesis-related primary metabolites across all candidate strains, identifying compounds that were efficiently synthesized (high PM values) and those likely requiring external supplementation (low PM values). Hierarchical clustering revealed substantial PM variability among the six strains (Fig. 5c). For instance, metal ions had consistently low PM values due to their external sourcing, while significant differences in cell wall and membrane components were observed between gram-negative strains (*A. brasilense* K, *P. stutzeri* G, and *P. fluorescens* J) and gram-positive strains (*C. cellulans* E, *B. velezensis* SQR9, and *B. megaterium* L). Amino acids are typically self-synthesized, and this explained their high PM values. However, *C. cellulans* E showed significantly lower PM value for asparagine, and *A. brasilense* K, for spermidine, tryptophan, isoleucine and asparagine. Further, variations in PM were noted in the vitamin B$_{12}$ precursors corrin and calomide, the vitamin K$_2$ derivatives menaquinone 8 and 2-demethylmenaquinone 8, and some membrane lipids, particularly lower in *A. brasilense* K. This could mean that these compounds are potentially used as units of metabolic exchange.

To verify whether these potential metabolites facilitate interactions in stable communities centered around narrow-spectrum resource-utilizing strains (that is, *C. cellulans* E and *P. stutzeri* G), we conducted metabolite supplementation assays (with the metabolites added at concentrations of 1 mM, 2 mM, and 5 mM) and evaluated their potential to mitigate population declines in SynComs with the two key strains (*C. cellulans* E and *P. stutzeri* G) dropped out. The tested

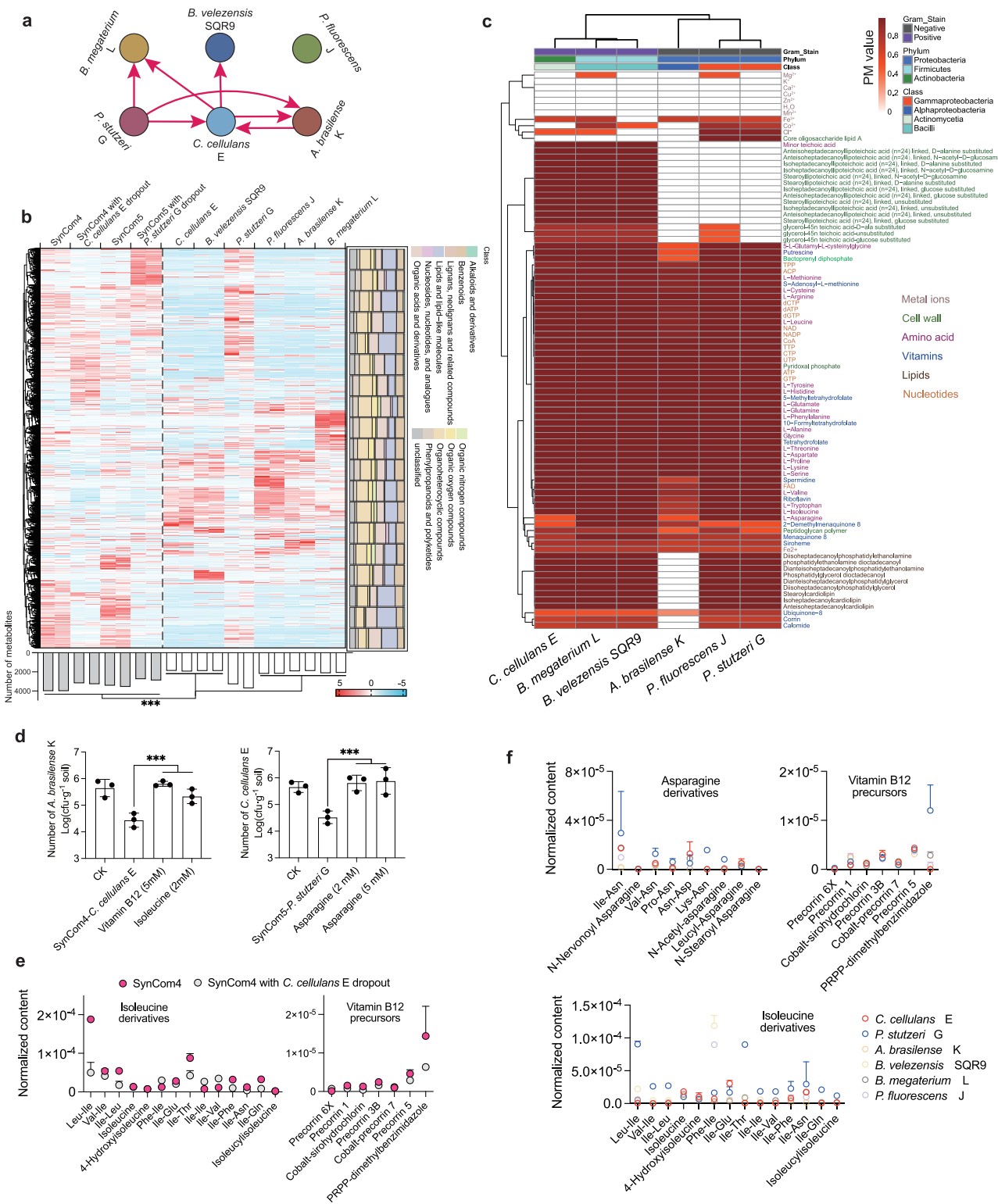

metabolites include galactose, tryptophan, isoleucine, serine, asparagine, vitamin $B_{12}$, and vitamin $K_2$. In SynCom5, supplementation with asparagine (2 mM or 5 mM) partially restored the *C. cellulans* E population following dropout of *P. stutzeri* G (Fig. 5d). In SynCom4, both vitamin $B_{12}$ (5 mM) and isoleucine (2 mM) fully restored the *A. brasilense* K population after *C. cellulans* E dropout (Fig. 5d). The other tested compounds had no significant effect on community composition (Supplementary Fig. 4). In addition, extracellular metabolite profiling revealed that the dropout of *C. cellulans* E from SynCom4 caused

a significant reduction in isoleucine and vitamin $B_{12}$ derivatives or precursors ($p < 0.01$, Fig. 5e). Asparagine levels are similar in SynCom5 with or without *P. stutzeri* G, but *P. stutzeri* G exhibited the strongest secretion (1.54–11.58 times than other strains and communities, $p < 0.01$, Fig. 5f), suggesting its potential role in asparagine supply in SynCom5. While other metabolites may also contribute to stability, metabolomic and metabolic modeling analyses support that *C. cellulans* E and *P. stutzeri* G stabilize SynCom4 and SynCom5 by supplying key metabolites such as vitamin B12, isoleucine, and asparagine.

**Fig. 5 | Interspecies interactions and validation of key metabolites in the synthesized communities.** All six plant-beneficial strains were cultured individually in artificial root exudate medium, and their post-fermentation supernatants were used to culture other five strains (**a**), with red arrows indicating positive growth support. Untargeted metabolomics identified 5,685 extracellular metabolites across nine chemical classes from individual strains, SynCom4, SynCom4 with *C. cellulans* E dropout, SynCom5, and SynCom5 with *P. stutzeri* G dropout after 48 h of fermentation on RE medium. A heatmap show metabolite clustering across samples (**b**), with red/blue indicating high/low abundance, respectively. Producibility metrics (PM, range 0–1) for 88 biomass-related primary metabolites were calculated for each strain using GMM-based simulations and hierarchically clustered (**c**).

Some unusual biosynthesis, such as $Mg^{2+}$, $Co^{2+}$, $Cl^-$, $Fe^{3+}$, and $Fe^{2+}$, can be explained based on their presence in larger compounds, such as porphyrins and vitamins. Predicted exchangeable key metabolites, including galactose, tryptophan, isoleucine, serine, asparagine, vitamin B12, and vitamin K2, were supplemented into dropout communities. Effects of asparagine on SynCom5 with *P. stutzeri* G dropout and vitamin B12/isoleucine on SynCom4 with *C. cellulans* dropout are shown (**d**). Metabolomic profiling revealed strain-level (**e**) and community-level (**f**) differences in the derivatives or precursors of asparagine, vitamin B12, and isoleucine. Error bars represent mean ± SD of biological replicates ($n = 3$). Statistical significance was determined by two-tailed $t$-test: $p < 0.001$ (***), $p < 0.01$ (**), and $p < 0.05$ (*). Source data are provided as a Source Data file.

## Multifunctional stable SynComs significantly promote tomato growth

Next, we investigated whether the stable SynComs designed here also had the ability to promote plant growth in comparison to the unstable SynComs or individual strains. To this end, pot experiments with tomato plants supplemented with various communities were conducted. It was revealed that among all the candidate strains, only *B. velezensis* SQR9 and *B. megaterium* L caused a significant increase in tomato growth (Fig. 6). In unstable communities, such as SynCom3-1 and SynCom3-2, no significant growth promotion was observed, even with the inclusion of *B. velezensis* SQR9 and *B. megaterium* L (Fig. 6). In contrast, SynCom4-1, SynCom4, and SynCom5 were associated with a significant increase in the shoot height and shoot dry weight of tomato plants compared to the control plants, with the stable SynCom4 and SynCom5 showing the strongest effects ($p < 0.01$; Fig. 6). Compared to the control, SynCom4 increased the plant height and dry weight by 58.6% and 85.6%, respectively, while SynCom5 showed 48.9% and 91.9% increase, respectively. These findings suggest that community stability and interactions are crucial to the targeted functionality of synthetic microbial consortia.

## Discussion

In this study, we applied metabolic modeling to guide the bottom-up construction of synthetic microbial communities in the plant rhizosphere. Our results demonstrate that strains with narrow-spectrum resource utilization enhanced community stability by increasing metabolic interaction potential (MIP) and reducing metabolic resource overlap (MRO). These findings provide a perspective that integrates resource utilization characteristics and metabolic modeling into the design of stable, multifunctional microbial consortia, which were validated for their effectiveness in promoting plant growth. This strategy appears to be promising for the future design of targeted microbial consortia.

Traditionally, stable microbial coexistence has been attributed to widespread competitive interactions[9,42], yet recent studies suggest that cooperation, particularly metabolic exchange, also plays a crucial role[17,43]. For instance, the reductionist hypothesis proposes that stable multispecies communities require that every pair of species can coexist independently[21], emphasizing pairwise compatibility. In contrast, other studies argue that coexistence emerges from higher-order interactions within the community context, such as cross-feeding, even when species pairs are competitively exclusive[44]. Machado et al. provided a broader view by showing that coexistence communities can be driven by either competition or cooperation[31]. Cooperative groups tend to possess fewer metabolic genes and stronger metabolic interactions, with higher abundance and broader ecological adaptability. These perspectives offer valuable guidance for synthetic community construction, and competition and cooperation are, to some extent, distinct strategies for microbial resource acquisition.

In plant rhizosphere, we observed significant variation in resource utilization strategies among bacterial strains. It is conceivable that NSR strains are more likely to secrete specific metabolites that enhance metabolic exchange, thereby playing a key role in niche differentiation,

reducing competition, and promoting cooperation, as supported by GMM analysis showing that NSR strains exhibit higher MIP scores and lower MRO scores. Representative strains like *P. stutzeri* G, *A. brasilense* K, and *C. cellulans* E formed dense cross-feeding networks, indicating evolved cooperative strategies. In contrast, BSR strains, such as *B. velezensis* SQR9 and *P. fluorescens* J showed low dependency on other members and maintained stable abundance under dropout conditions, reflecting a competitive, self-sufficient lifestyle. However, our pot experiments revealed that cooperative communities enriched with NSR strains outperformed non-cooperative ones in promoting plant growth, even when the latter included well-known growth-promoting strains such as *B. velezensis* SQR9 and *B. megaterium* L. This emphasizes that metabolic cooperation, rather than individual strain potential, is a key driver of community functionality.

Amino acids are commonly considered the major exchanged metabolites in cooperative communities[31]. However, our refined results suggest that bacteria may preferentially exchange amino acid derivatives, particularly oligopeptides, which exist abundantly and diversely in extracellular secretions. Especially in non-absolute auxotrophic communities, these functionally active metabolites likely follow complex and dynamic exchange patterns influenced by molecular form, bioavailability, and contextual efficacy.

Metabolic modeling offers clear advantages in designing stable, multifunctional communities, yet some limitations remain. Genomic predictions of resource utilization width correlated well with experimental measurements. when experimental values were used as the reference, both experimental and their corresponding predicted values were significantly correlated with community MIP or MRO scores. However, using predicted values as the reference led to a loss of this pattern ($R^2 < 0.01$, not shown). This discrepancy arises from error between predicted and experimental values, and when using predicted values to construct MIP/MRO correlations (the second type of error), the cumulative error leads to the loss of the observed pattern. This clearly highlights the importance of calibrating metabolic models with real values. Studying large microbial communities is a significant challenge due to the extensive workload. We believe that microbial genome-scale metabolic modeling, combined with high-throughput calibration, offers a promising strategy.

In conclusion, cooperative metabolic interactions are essential for constructing stable and functional microbial communities. Narrow-spectrum resource-utilizing strains not only promote coexistence through niche partitioning but also enhances metabolic cooperation and ecological function. Therefore, when designing synthetic microbial consortia, especially for plant-associated habitats, it is crucial to integrate metabolic modeling and prioritize strains that contribute to community-level cooperation and stability.

## Methods
### Strains and growth conditions

The six strains used in this study were originally isolated from the plant rhizosphere and were obtained from the Agricultural Culture Collection of China (ACCC) and China General Microbiological Culture Collection Center (CGMCC): *C. cellulans* E (ACCC accession number

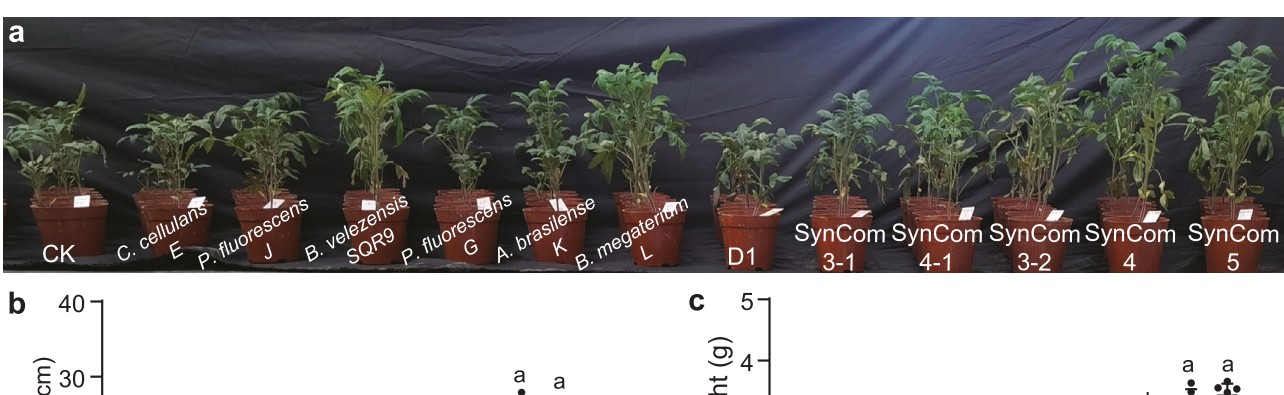

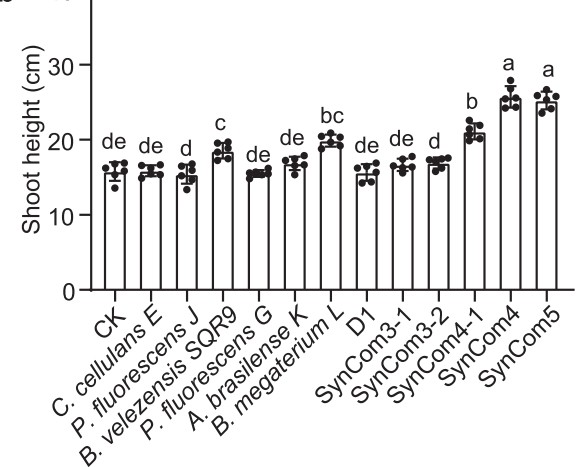

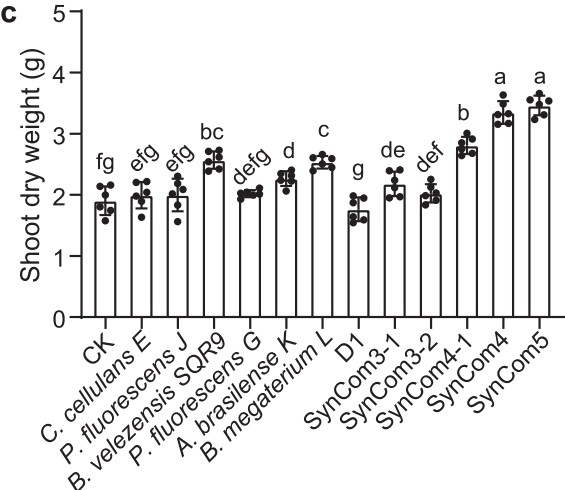

**Fig. 6 | Plant growth-promoting effects of different synthetic communities.**
**a** Photograph of 6-week-old tomato plants grown in soil inoculated with individual strains or synthetic communities. **b** Shoot height. **c** Shoot dry weight. CK represents the non-inoculated control, and D1 is a heat-inactivated mixture of the six strains. Bars represent the mean ± s.d. (*n* = 6). Bars with different letters indicate significant differences (*p* < 0.05, one-way ANOVA, two-sided), while those sharing at least one letter are not. Error bars represent mean ± SD of biological replicates (*n* = 6). Source data are provided as a Source Data file.

01019), *P. stutzeri* G (ACCC accession number 06513), *P. fluorescens* J (ACCC accession number 04268), *B. megaterium* L (ACCC accession number 10010), *A. brasilense* K (CGMCC accession number 1.10379), and *B. velezensis* SQR9 (CGMCC accession number 5808). The starting inoculum was cultured at 30 °C in low-salt Luria-Bertani (LLB) medium (10 g·L⁻¹ peptone; 5 g·L⁻¹ yeast extract; 3 g·L⁻¹ NaCl), centrifuged at $5000 \times g$ for 2 min, and resuspended in 0.9% (w/v) NaCl to obtain a concentration of $1 \times 10^7$ cfu·mL⁻¹. The multi-species inoculum was prepared by mixing equal volumes of each single-species inoculum.

Each strain was inoculated into the artificial root exudate (RE) medium at a concentration of $1 \times 10^6$ cfu·mL⁻¹ and incubated at 30 °C, 170 rpm for 48 h. Cells were then removed by centrifugation, and post-fermentation supernatants were filter-sterilized. To assess cross-utilization, each strain was inoculated into the post-fermentation supernatants of the other strains at $1 \times 10^6$ cfu·mL⁻¹ and incubated under the same conditions. For the control setting, each strain was also inoculated into its own post-fermentation supernatant to exclude the influence of residual nutrients. Microbial growth was measured at OD₆₀₀, with six replicates for each treatment. RE medium consisted of M9 inorganic salt medium (Sigma-Aldrich M6030) supplemented with sugars (glucose, fructose, sucrose, arabinose, xylose, maltose, and galactose, each at a concentration of 1 mmol·L⁻¹), organic acids (succinic, malic, tartaric, oxalic, citric, pyruvic, and malonic, each at a concentration of 0.5 mmol·L⁻¹) and amino acids (glutamic, alanine, threonine, serine, valine, glycine, histidine, lysine, arginine, leucine, and phenylalanine; each at a concentration of 0.063 mmol·L⁻¹).

### Functional analysis of plant-beneficial strains
The strains were tested for plant-beneficial traits including nitrogenase activity, siderophore production, phosphate solubilization, and IAA production. Nitrogenase activity was measured as described by Hunt et al.[45]. Briefly, strains were cultured in LB broth (37 °C, 170 rpm,

overnight), washed, and resuspended in Ashby's medium. A 1% inoculum was transferred into 10 mL semi-solid Ashby's medium in serum bottles, which were sealed and purged with argon gas to ensure anaerobic conditions. Acetylene (10%, v/v) was injected, and bottles were incubated at 28 °C for 72 h. Ethylene production in 250 μL gas samples was quantified by gas chromatography. Siderophore production was tested using the chrome azurol S (CAS) agar method, as outlined by Gu et al.[46]. Briefly, strains (1% inoculum) were cultured in MKB medium (30 °C, 170 rpm, 2 d). After centrifugation (10,000 × g, 5 min), supernatants were mixed 1:1 with CAS reagent, incubated at room temperature, and absorbance at 630 nm was measured. Siderophore production using the follow equation:

$$\text{Siderophore production} = 1 - (A_s / A_r) \tag{1}$$

where $A_s$ is the sample absorbance and $A_r$ is the absorbance of the uninoculated control. Phosphate solubilization was evaluated on NBRIP agar containing calcium phytate or $Ca_3(PO_4)_2$, according to Hu et al.[47]. Briefly, strains were cultured in NBRIP broth (28 °C, 170 rpm, 5 d), and supernatants were collected by centrifugation (10,000 × g, 5 min). Samples were diluted, pH-indicated with methyl orange, reacted with molybdenum-antimony reagent, and absorbance at 700 nm was measured. IAA production were measured following Ahmad et al.[48]. Briefly, Strains were grown in tryptophan-supplemented Landy medium (25 °C, 140 rpm, dark, 72 h). Supernatants (2 mL) were mixed with two drops of orthophosphoric acid and 4 mL Salkowski reagent (50 mL of 35% perchloric acid with 1 mL of 0.5 M FeCl₃), and absorbance was measured at 530 nm.

### Resource utilization profile assay
The growth of six plant-beneficial strains on 58 rhizosphere carbon sources was evaluated using the Biolog Phenotype MicroArray system,

following the manufacturer's protocol. Briefly, cells were inoculated onto Biolog plates PM1 and PM2 and incubated at 30 °C in an Omnilog incubator-reader (BIOLOG, USA) for 48 h. Color changes from colorless to purple, indicative of cellular respiration through the reduction of tetrazolium-based dyes, was monitored every 30 min. Due to a supply shortage of the Biolog plates in the later stages of the study, resource utilization of the 25 additional laboratory-stored rhizosphere strains was assessed using an alternative approach adapted from Schäfer et al.[40]. Briefly, strains were grown on M9 agar medium supplemented with each of 41 available carbon sources (Supplementary Data 1), and carbon-free M9 agar medium served as the negative control to determine resource utilization.

### Calculation of resource utilization width and overlap

Levins' niche width index[49] was applied to quantify resource utilization width, as depicted in the equation below.

$$W = \frac{1}{\sum \left(P_i^2\right)} \quad (2)$$

$W$ represents the resource utilization width, and $P_i$ is the utilization rate of resource $i$ for a particular species. An OmniLog value greater than 50 was set as the threshold to determine whether a strain could utilize a given resource. Higher $W$ values indicate broader resource utilization width. In validation experiments using 224 phyllosphere strains and 25 laboratory-stored rhizosphere strains, resource utilization width was simplified as the total number of substrates that supported growth.

Pianka's overlap index[50] was applied to quantify resource utilization overlap between species, and was calculated using the following equation.

$$O_{jk} = \frac{\sum \left(P_{ik} \times P_{ij}\right)}{\sqrt{\sum P_{ik}^2 \times \sum P_{ij}^2}} \quad (3)$$

$O_{jk}$ is the resource utilization overlap index between species $k$ and $j$. $P_{ik}$ and $P_{ij}$ represent the proportions of resource $i$ used by species $k$ and $j$, respectively. The summation spans all resource categories. This index ranges from 0 (no overlap) to 1 (complete overlap), indicating the degree of similarity in resource utilization between two species.

### Genome sequencing

An Illumina Hiseq 4000 sequencer combined with third-generation sequencing technology (Pacific Biosciences, Menlo Park, CA, USA) was used for genome sequencing, which was conducted by Gene Denovo Biotechnology Co. (Guangzhou, China). Long reads from single molecular real-time sequencing on the PacBio platform were used for de novo assembly, while raw data from the Illumina platform were utilized for error correction to enhance the genome assembly quality. The raw sequencing data and assembled genomes have been deposited in the database of the National Center for Biotechnology Information (NCBI) under the BioProject accession numbers PRJNA666038, PRJNA647852, PRJNA647624, PRJNA647632, and PRJNA647635. Data for *B. velezensis* SQR9 were sourced from NCBI under BioProject accession number PRJNA227504.

### Model construction and interaction simulation

Genome-scale metabolic models for all strains were constructed using CarveMe (v1.5.0) with default parameters[51]. The draft models were refined based on carbon source utilization profiles obtained from phenotype microarrays using the NICEgame software[52], the BIGG database[53], and thermodynamic reaction constraints[54]. During CarveMe construction, exchange reactions for intra- and extracellular metabolites are assigned to each model based on its annotated

metabolic gene content. Accordingly, predicted carbon utilization width was defined as the number of exchange reactions involving carbon-containing organic compounds. Community simulations were performed with SMETANA[30] using default parameters, where models of different communities were input to simulate MIP and MRO. Phylogenetic distances between bacterial community members were extracted from a phylogenetic tree constructed using GTDB-Tk v2.4.0[55].

### Assessment of the stability of SynComs in the tomato rhizosphere

Natural soil samples were sterilized by γ-irradiation, and surface-sterilized tomato seeds were germinated on moist filter paper before they were transplanted into vermiculite with the 1/4 MS medium. The plants were grown in sterile soil until the two-leaf stage. The microbial communities SynCom3-1, SynCom3-2, SynCom4-1, SynCom4 and SynCom5 were inoculated into the soil at a concentration of $10^6$ cfu·g$^{-1}$ per strain. Rhizosphere soil samples were collected at 0 h, 6 h, 24 h, 48 h and 60 h intervals. DNA was extracted from the rhizosphere soil using the DNeasy PowerSoil Kit (QIAGEN, Germany), following the manufacturer's instructions. Each treatment included six replicates. Strain dropout experiments were conducted using the same methods.

Whole-genome sequences of the six strains were aligned using BLAST (2.5.0+) to identify strain-specific regions. Based on these sequences, primers for qPCR were designed using Primer Premier 6 (Premier Biosoft, USA), and their specificity was validated through conventional PCR (Supplementary Table 1). Quantification of strain abundance in rhizosphere samples was performed using an Applied Biosystems real-time PCR system, with each treatment comprising six biological replicates.

### Non-targeted metabolomics analysis

Strains were pre-cultured in LLB medium, harvested by centrifugation (5000 × $g$), and washed twice with sterile 0.9% (w/v) NaCl. Cell suspensions were adjusted to $1 \times 10^7$ cfu·mL$^{-1}$ and inoculated into RE medium for individual strains, SynCom4, SynCom5, SynCom4 with *C. cellulans* E dropout, and SynCom5 with *P. stutzeri* G dropout. All cultures were standardized to $1 \times 10^6$ cfu·mL$^{-1}$ total inoculum, with equal proportions per strain in mixed communities. After 48 h incubation at 30 °C and 170 rpm, 2 mL of culture was centrifuged, and the supernatant filtered (0.22 μm). Cell-free supernatants were flash-frozen in liquid nitrogen and stored at −80 °C for metabolomic analysis.

Metabolomic identification was commissioned to BioTree Biotech Co. (Shanghai, China). Briefly, metabolites were extracted by mixed 50 μL of sample with 200 μL of methanol:acetonitrile (1:1, v/v) containing internal standards. After vortexing (750 rpm, 5 min), settling (5 min), homogenization (35 Hz, 4 min), and sonication (10 min, 4 °C), samples were incubated at −40 °C for 1 h to precipitate proteins, then centrifuged (12,000 × $g$, 15 min, 4 °C). Supernatants were collected for LC-MS/MS analysis using a Vanquish UHPLC system coupled to an Orbitrap Exploris 120 mass spectrometer (Thermo Fisher Scientific, USA). Polar and non-polar metabolites were separated on BEH Amide and Kinetex C18 columns, respectively. MS operated in IDA mode, with resolutions of 60,000 (MS) and 15,000 (MS/MS), and SNCE at 20/30/40. Raw data were processed using an in-house R pipeline based on XCMS. Metabolite annotation was performed utilizing a commercial in-house MS2 database (BiotreeDB v3.0)[56]. A total of 355,333 features were detected, with 222,588 retained after RSD filtering. Missing values were imputed by half the minimum value, and total ion current normalization was applied.

### Producibility metric calculation

To quantify the metabolic robustness of target metabolite production under variable conditions, we followed the Producibility

metric (PM) framework proposed by Bernstein et al.[57]. Briefly, all exchangeable metabolites in the model were treated as potential environmental inputs, each assigned an input probability ($P_{in}$) representing its likelihood of availability. Random sampling based on $P_{in}$ generated multiple hypothetical environments, and flux balance analysis (FBA) was used to determine whether the target metabolite could be synthesized (output probability, $P_{out}$). By varying $P_{in}$ from 0 to 1, a producibility curve describing $P_{out}$ as a function of $P_{in}$ was obtained. The $P_{in}$ value at which $P_{out}$ reaches 0.5 ($P_{in,0.5}$) was identified, and PM was calculated as

$$PM = 1 - P_{in,0.5} \qquad (4)$$

with higher PM values indicating greater robustness across environmental conditions.

## Plant growth-promoting effects assay

The plant growth-promoting experiment was conducted at the BaiMa Experimental Base in Nanjing, China. The soil used for the experiments had the following properties: pH, 7.7; organic matter, 3.4 g·kg⁻¹; available nitrogen (N), 130 mg·kg⁻¹; available phosphorus (P), 19.5 mg·kg⁻¹; available potassium (K), 158 mg·kg⁻¹; total nitrogen (N), 0.9 g·kg⁻¹; total phosphorus (P), 0.3 g·kg⁻¹, and total potassium (K), 21.1 g·kg⁻¹. The experimental treatments included a non-inoculated control (CK), individual strain inoculations, a heat-inactivated mixture of the six strains, and synthetic communities SynCom3-1, SynCom3-2, SynCom4-1, SynCom4, and SynCom5. All live bacterial treatments, including single strains and synthetic communities, were inoculated at a final density of 10⁶ cells·g⁻¹ soil. For community treatments, strains were mixed in equal proportions. Plants were grown for 30 days at 30 °C with a 16 h light/8 h dark cycle. Each treatment was performed in six replicates.

## Reporting summary

Further information on research design is available in the Nature Portfolio Reporting Summary linked to this article.

## Data availability

The sequencing data and assembled genomes data can be accessed from the NCBI database under accession PRJNA666038, PRJNA647852, PRJNA647624, PRJNA647632, PRJNA647635, PRJNA1255786, and PRJNA1256194. The raw metabolomics files were uploaded to the Metabolights repository under the identifier MTBLS12572. Source data are provided with this paper.

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

## Acknowledgements

This work was financially supported by the National Key Research and Development Program (2022YFF1001804 to R.Z., 2024YFD1701600 to Y.M., and 2021YFD1900300 to W.X.), the General Program of National Natural Science Foundation of China (42477126 to Y.M.) and the Jiangsu Provincial Natural Science General Program (BK20231472 to Y.M.).

## Author contributions

Y.M. and R.Z. designed the study. W.W., Y.X., P.Z., M.Z. and S.H. performed the experimental work. W.W. and X.S. performed the metabolic modeling. Y.M. and W.W. performed the data analysis. Y.M., W.W., R.Z., Z.X., N.Z., W.X. and Q.S. discussed the results and drafted the manuscript. Y.M. wrote the final version of the manuscript. All authors read and approved the final manuscript.

## Competing interests

The authors declare no competing interests.
