## [Peer Review file · Nature Communications]

Narrow-spectrum resource-utilizing bacteria drive the stability of synthetic communities through enhancing metabolic interactions

Corresponding Author: Dr Youzhi Miao

Version 0:

Reviewer comments:

Reviewer #1

(Remarks to the Author)

Authors established a strategy to constructing a synthetic microbial community that can enhance plant growth. In this study, six bacterial strains were selected as candidate strains based on their plant growth promoting traits. Metabolic interaction was considered as the strategy of constructing stable and multifunctional bacterial community and it was verified through genome-based metabolic models and lab experiment. However, there are still some issues need to be addressed.

1. Lacking of background information and relevant data on selection of the six candidate strains. Selection of the strains or initial microbial communities is essential of this research. Please provide more information on how many strains were isolated and according to what analyses the six candidate strains were selected.
2. Have you proved this strategy with other bacterial strains or even with a random communities? If this strategy only works on these 6 strains, which will clearly influence on the novelty and limit the potential of future application.
3. Knockout a strain from the SynCom is the approach to verify the stability of the communities. The term "knockout" caused confusion. "knockout" typically refers to the loss of gene function at the genetic level, which is not the case in this study. I suggest to replace "knockout" with "dropout".
4. I am confused with the origin of the metabolites that authors used to clarify the interaction. Authors firstly showed the significance about post-fermentation supernatant, and I assume these metabolites (asparagine, vitamin B12, isoleucine) from candidate strains. Authors then calculated PM with all candidate strains but did not offer the solid evidence that these metabolites come from the bacteria strains. It is essential to prove these metabolites (asparagine, vitamin B12, isoleucine) were produced by *C. cellulans* E and *P. stutzeri* G. Please prove more data here.
5. It is hard to understand the figure 2c, 2e, 3a, 3b, 3c. Visualization of these figures need to be improved. Additionally, the legends are incomplete in figure 3b and 3c. Please correct these.

Reviewer #2

(Remarks to the Author)

Wang et al describe the characterisation of nutrient utilisation profiles of root-associated bacteria, their assembly into communities and ultimately their effect on tomato plant growth. Different bacteria are found to lie on different points on the generalist – specialist scale, as measured by their 'resource-utilizing width' of different carbon/nitrogen sources in vitro. Metabolic models fitted to the genomes and the in vitro nutrient utilisation data are able to recapitulate the expected behaviour of generalists vs specialists in communities of different sizes, ie communities with largely broad resource-utilising width also have larger metabolic resource overlap (MRO) scores. Communities containing the specialists *C. cellulans* and *P. stutzeri* are found to colonise plant roots more effectively. Supernatant from these species promotes the growth of other species in vitro, suggesting that these communities are cooperative. The authors then make an effort to identify putative exchanged metabolites and identify some amino acids and vitamins. Finally, these cooperative communities substantially promote tomato growth, potentially due to the increased abundance of *Bacillus* sp.

This is a well-written paper that makes effective use of different in vitro, in silico and in planta techniques. The authors provide mechanistic depth to their observations, although I have some concerns about the 'producibility metric' used to

identify the cross-fed metabolites. While the notion that communities of specialists are more cooperative than communities of generalists is not new (Machado et al 2021), the nuances within root-associated bacteria are interesting. The strategies and observations shown here might indeed be useful to construct communities for agricultural applications.

1. What is the producibility metric and which tool was used to compute it? The Methods contain no detail on this. Some species seem to be able to 'produce' metal ions, which doesn't make sense. Please elaborate.
2. While the metabolite supplementation results are supportive, identity of exchanged metabolites would need confirmation using direct metabolite measurements in the community supernatants.
3. Explain MIP and MRO at the first use.
4. Show data points in the bar plots.
5. Figures:
 - Fig 2A – why some substrates are in red?
 - Fig 2B&C – legend indicates that this plot contains 57 datapoints, I can spot much fewer. Please consider changing the visualisation to make the individual data points more visible.
 - Fig 2C – These seem to me like 'sanity check' results rather than revealing new biological information, especially since the metabolic models were refined using the Biolog data.
 - Fig S2 – please improve the visualisation (e.g. use thin lines without markers for every data point, use colors to distinguish)
 - Fig 5bc – I do not understand the small letter written on top of the bars, please explain better.

Reviewer #3

(Remarks to the Author)

Version 1:

Reviewer comments:

Reviewer #1

(Remarks to the Author)

Authors addressed all issues nicely and have substantially revised the manuscript. I do not have further criticisms on the revised version.

Reviewer #2

(Remarks to the Author)

The authors have addressed most of my comments. There are two outstanding critique points:

- 1a. The new metabolomics data is not publically available.
- 1b. It is unclear from the Methods what level of evidence was used for metabolite annotation (retention time? &/ MS1? &/M S2?)
2. regarding my previous comment, "Fig 2C – These seem to me like 'sanity check' results rather than revealing new biological information, especially since the metabolic models were refined using the BiOLOG data.": The authors have misunderstood the point. The point is that if BioLOG data is used to build models, it is a circular argument to use the same data to validate.

Reviewer #3

(Remarks to the Author)

Version 2:

Reviewer comments:

Reviewer #2

(Remarks to the Author)

I am afraid that the authors have missed again the point on the use of BIOLOG data for model building/validation. There is absolutely nothing wrong in using BIOLOG to build the model. In fact, it is highly recommended and should be used. My point is that consistency of the resulting models with the data used to build the models should not be phrased as "validation". I

suggest only to rephrase along these lines, not to revise figures using models. that do not use BIOLOG data.

Version 3:

Reviewer comments:

Reviewer #2

(Remarks to the Author)

The authors have addressed my remaining concern. Congratulations on a nice study.

Reviewer #1 (Remarks to the Author):

Authors established a strategy to constructing a synthetic microbial community that can enhance plant growth. In this study, six bacterial strains were selected as candidate strains based on their plant growth promoting traits. Metabolic interaction was considered as the strategy of constructing stable and multifunctional bacterial community and it was verified through genome-based metabolic models and lab experiment. However, there are still some issues need to be addressed.

Response: Thank you for your valuable feedback. We have carefully considered your comments and made comprehensive revisions to the manuscript.

1. Lacking of background information and relevant data on selection of the six candidate strains. Selection of the strains or initial microbial communities is essential of this research. Please provide more information on how many strains were isolated and according to what analyses the six candidate strains were selected.

Response: Thank you for your valuable comment. We agree that clearly describing the selection strategy for the six candidate strains at the outset is important. As we know, current approaches for constructing synthetic communities typically involve simplifying complex native communities based on species abundance, association networks, or co-occurrence patterns. As mentioned in the Introduction, strategies for assembling stable communities by directly selecting strains from known isolates are still under explored, which is the focus of this study. Therefore, we adopted a function-driven approach to select six candidate strains, each representing a well-studied species with recognized plant-beneficial functions. Although straightforward, this strategy aligns closely with the concept of de novo community assembly. We have clarified this rationale in the revised manuscript (**Lines 113-119**). Thank you again for your suggestion.

2. Have you proved this strategy with other bacterial strains or even with a random community? If this strategy only works on these 6 strains, which will clearly influence on the novelty and limit the potential of future application.

Response: Thank you for your valuable suggestion. Demonstrating the generality of the principle that "narrow resource-utilizing strains promote community stability through enhanced metabolic interactions" is crucial for the broader applicability of this

strategy. To support this, we analyzed data from Schäfer *et al.* (Science, 2023), which included 224 bacterial models from *Arabidopsis thaliana* phyllosphere microbiota and their utilization of 45 carbon sources. The result showed a near-normal distribution of bacterial resource utilization widths. Similarly, simulation analysis of 3001 rhizosphere bacterial genomes published by Dai *et al.* (Cell, 2025) confirmed this trend. These findings suggest that in plant-associated microbiota, beyond the majority of species with moderate resource utilization, a stable subset tends to adopt ecological strategies involving either markedly reduced or expanded resource use, corresponding to narrow-spectrum and broad-spectrum resource-utilizing strains, respectively. Further analysis revealed significant negative correlations between bacterial resource utilization widths and community MIP scores, as well as positive correlations with MRO scores. Importantly, these correlations were significantly strengthened as community size or evolutionary distance increased. We also tested 25 rhizosphere bacterial strains stored in our lab, which yielded similar results. Overall, our analysis supports that NSR strain-driven cooperation represents a common pattern in plant-associated bacteria. The detailed results are provided in **lines 182-224** of the revised manuscript. Thank you again for your suggestion, which has greatly enriched our study.

3. Knockout a strain from the SynCom is the approach to verify the stability of the communities. The term “knockout” caused confusion. “knockout” typically refers to the loss of gene function at the genetic level, which is not the case in this study. I suggest to replace “knockout” with “dropout”.

Response: Thank you for the suggestion. We have replaced “knockout” with “dropout” throughout the manuscript to avoid confusion with gene-level manipulations.

4. I am confused with the origin of the metabolites that authors used to clarify the interaction. Authors firstly showed the significance about post-fermentation supernatant, and I assume these metabolites (asparagine, vitamin B12, isoleucine) from candidate strains. Authors then calculated PM with all candidate strains but did not offer the solid evidence that these metabolites come from the bacteria strains. It is essential to prove these metabolites (asparagine, vitamin B12, isoleucine) were produced by *C. cellulans* E and *P. stutzeri* G. Please prove more data here.

Response: Thank you for pointing out the lack of evidence in the original manuscript regarding the production of these metabolites by *C. cellulans* E and *P. stutzeri* G. We have addressed this in the revised version by adding metabolomic identification and analysis, which better supports this point. Given the complexity of metabolic exchanges

in non-absolute auxotrophic systems, we have revised the logical structure of our results as follows:

Post-fermentation experiments confirmed the metabolic support provided by NRS strains (*C. cellulans* E and *P. stutzeri* G) to other strains. Metabolomic profiling revealed distinct extracellular metabolite distributions across six individual strains, suggesting metabolic exchange potential. Compared to single-strain cultures, the extracellular metabolite distribution in SynCom4 and SynCom5 was more diverse and divergent, indicating that community assembly facilitated complete metabolite exchange and metabolic reorganization. Removal of *C. cellulans* E from SynCom4 or *P. stutzeri* G from SynCom5 caused substantial disruption in metabolite profiles, highlighting their importance in the community's metabolic network. Given the complexity of over 5600 metabolites identified, including numerous secondary metabolites, we further used metabolic modeling to predict the synthesis potential of 88 biomass-related primary metabolites, identifying growth-limiting compounds. Supplementation with the identified asparagine, vitamin B12, and isoleucine restored the abundance of related members after the removal of *C. cellulans* E and *P. stutzeri* G from communities. Further metabolomic analysis demonstrated that *C. cellulans* E and *P. stutzeri* G have a significant advantage in secreting asparagine, vitamin B12, isoleucine, and their precursors or derivatives. While other metabolites may also be exchanged, integrated metabolomic and metabolic modeling analyses confirmed that *C. cellulans* E and *P. stutzeri* G stabilize SynCom4 and SynCom5 by supplying essential metabolites such as vitamin B12, isoleucine, and asparagine. Detailed results and discussions are provided in **lines 276-330** and **lines 385-391** of the revised manuscript.

5. It is hard to understand the figure2c,2e,3a,3b,3c. Visualization of these figures need to be improved. Additionally, the legends are incomplete in figure 3b and 3c. Please correct these.

Response: We acknowledge that the previous images did not effectively present the data. Following your suggestion, we have made the necessary adjustments to both the images and legends. Thank you very much.

Reviewer #2 (Remarks to the Author):

Wang et al describe the characterization of nutrient utilization profiles of root-associated bacteria, their assembly into communities and ultimately their effect on tomato plant growth. Different bacteria are found to lie on different points on the

generalist – specialist scale, as measured by their ‘resource-utilizing width’ of different carbon/nitrogen sources in vitro. Metabolic models fitted to the genomes and the in vitro nutrient utilization data are able to recapitulate the expected behavior of generalists vs specialists in communities of different sizes, i.e. communities with largely broad resource-utilizing width also have larger metabolic resource overlap (MRO) scores. Communities containing the specialists *C. cellulans* and *P. stutzeri* are found to colonize plant roots more effectively. Supernatant from these species promotes the growth of other species in vitro, suggesting that these communities are cooperative. The authors then make an effort to identify putative exchanged metabolites and identify some amino acids and vitamins. Finally, these cooperative communities substantially promote tomato growth, potentially due to the increased abundance of *Bacillus* sp.

This is a well-written paper that makes effective use of different in vitro, in silico and in planta techniques. The authors provide mechanistic depth to their observations, although I have some concerns about the ‘producibility metric’ used to identify the cross-fed metabolites. While the notion that communities of specialists are more cooperative than communities of generalists is not new (Machado et al 2021), the nuances within root-associated bacteria are interesting. The strategies and observations shown here might indeed be useful to construct communities for agricultural applications.

Response: Machado *et al.* developed CarveMe software and revealed a polarization of coexisting microbial communities in ecological systems, driven respectively by competitive or cooperative metabolism. Cooperative groups tend to possess fewer metabolic genes, stronger metabolic interactions, higher abundance, and broader environmental adaptability. In this study, we focus on the plant’s second genome—the rhizosphere microbiome—to explore the construction strategies for multifunctional and stable microbial consortia, aiming to translate theoretical insights into practical applications and uncover potential challenges. Machado’s work provided an important conceptual reference for our study, and we have added the necessary discussion in **lines 364-369**. Thank you for your suggestions and support for this work, and we have carefully revised and improved the manuscript based on your advice.

1. What is the producibility metric and which tool was used to compute it? The Methods contain no detail on this. Some species seem to be able to ‘produce’ metal ions, which doesn’t make sense. Please elaborate.

Response: We apologize for the oversight and appreciate your reminder. A detailed

description of the producibility metric calculation is provided in **lines 541-552** of the revised manuscript, based on Bernstein *et al.* (2019). Briefly, all exchangeable metabolites in the model were treated as potential environmental inputs, each assigned an input probability (P_{in}) representing its availability. Random sampling based on P_{in} generated multiple hypothetical environments, and flux balance analysis (FBA) determined whether the target metabolite could be synthesized (output probability, P_{out}). By varying P_{in} from 0 to 1, a producibility curve describing P_{out} as a function of P_{in} was obtained. The P_{in} value at which P_{out} reaches 0.5 ($P_{in,0.5}$) was identified, and PM was calculated as $PM = 1 - P_{in,0.5}$, with higher PM values indicating greater robustness across environments. Given the complexity of community metabolic exchanges, this method was adapted to identify primary metabolites potentially involved in exchanges that could limit strain growth. Regarding the unusual biosynthesis of Mg^{2+} , Co^{2+} , Cl^- , Fe^{3+} , and Fe^{2+} , the original method article provides a direct explanation: these ions are present in larger compounds, such as porphyrins and vitamins. We have added this explanation in the Fig. 5 legend (**lines 754-756**) of the revised manuscript. Thank you very much.

2. While the metabolite supplementation results are supportive, identity of exchanged metabolites would need confirmation using direct metabolite measurements in the community supernatants.

Response: Thank you very much for your suggestion. In the revised manuscript, we have added an analysis of the extracellular metabolomes of individual strains and communities. The results indicate that *P. stutzeri* G and *C. cellulans* E have a significant advantage in secreting asparagine, vitamin B12, isoleucine, and their precursors or derivatives. Dropout of these two strains from communities caused a notable decrease in these compounds and substantial disruption in the metabolite profiles, further supporting the core role of *P. stutzeri* G and *C. cellulans* E in metabolic cross-feeding. Therefore, the combination of metabolic model predictions and metabolomics data effectively highlights the cross-feeding dynamics of NSR strain-centered stable communities. In microbial interaction systems that are not strictly auxotrophic, the relationships and forms of metabolic exchanges may be more complex, and we do not exclude the presence of other equivalent exchange compounds. Detailed results are provided in **lines 276-330**, with further discussion in **lines 385-391**.

3. Explain MIP and MRO at the first use.

Response: We apologize for the oversight. Definitions of MIP and MRO have been

added at their first mention (**Lines 163-164**). Thank you for your suggestion.

4. Show data points in the bar plots.

Response: Thank you for your suggestion. We have added data points to the bar plots in Fig. 1 and Fig. S4.

5. Figures:

Fig 2A – why some substrates are in red?

Response: The red markings indicate substrates utilized by all six strains. We apologize for not noting this in the figure legend. Since this information is not central, we have removed the red highlighting in the revised manuscript. Thank you.

Fig 2B&C – legend indicates that this plot contains 57 datapoints, I can spot much fewer. Please consider changing the visualization to make the individual data points more visible.

Response: Thank you for your suggestion. The 57 community combinations refer to all possible combinations of 6 functional strains with 2-6 members. Fig. 2c includes paired combinations, while Fig. 2e shows combinations with 3-6 members. We apologize for the confusion in the original figure and have revised the presentation for clarity.

Fig 2C – These seem to me like ‘sanity check’ results rather than revealing new biological information, especially since the metabolic models were refined using the BiOLOG data.

Response: Your question is highly insightful. The patterns shown in Fig. 2c and Fig. 2e may seem overly perfect, and we initially shared the same doubt, although non-BIOLOG-refined models exhibited similar patterns (not shown). However, these are based on only six strains, and further validation of generalizability is needed.

To this end, we collected 224 bacterial metabolic models from Schäfer *et al.* (Science, 2023) on plant phyllosphere microbiota and their utilization of 45 carbon sources. The results confirmed significant negative correlation between bacterial resource utilization widths and community MIP scores ($R^2 = 0.2749$, $p < 0.0001$), and positive correlation with community MRO scores ($R^2 = 0.3915$, $p < 0.0001$). Notably, these correlations were strengthened as community members’ evolutionary distances or community size increased. Additionally, we sequenced the genomes of 25 rhizobacteria strains and constructed metabolic models (gap-filling only with M9 medium), along with testing 41 carbon source utilizations. The analysis also showed similar significant patterns.

These results suggest that the pattern of narrow-spectrum resource-utilizing strains enhancing community interactions is a general phenomenon in plant-associated bacteria.

In fact, genomic information can effectively predict bacteria's potential resource utilization width, which is highly positively correlated with the experimental resource utilization width ($R^2 = 0.4057$, $p < 0.0001$). It is worth noting that when using experimental resource utilization width as the reference, both experimental resource utilization widths and their corresponding predicted values were significantly correlated with community MIP or MRO scores. However, when using predicted resource utilization width as the reference, this pattern nearly disappears ($R^2 < 0.01$, not shown). This discrepancy arises from error between predicted and experimental resource utilization widths, and when using predicted values to construct MIP/MRO correlations (the second type of error), the cumulative error leads to the loss of the observed pattern. This result clearly illustrates the importance of calibrating with real values when revealing general patterns through metabolic models and also indicates that the correlation between community resource utilization width and MIP/MRO has biological significance.

In conclusion, studying large microbial communities is a significant challenge due to the extensive workload. We believe that microbial genomic metabolic modeling is a promising strategy, and incorporating high-throughput measurements for calibration is an attractive direction. Detailed results (**lines 182-224**) and discussions (**lines 392-404**) are provided in the revised manuscript, thank you very much for your valuable suggestion.

Fig S2 – please improve the visualization (e.g. use thin lines without markers for every data point, use colors to distinguish)

Response: Thank you very much for your suggestion. We have revised the image presentation accordingly.

Fig 5bc – I do not understand the small letter written on top of the bars, please explain better.

Response: We apologize for the confusion. These numbers represent statistical differences, and we have provided corresponding explanations in the figure legend. Thank you.

Reviewer #3 (Remarks to the Author):

Response: Thank you for your valuable contribution to the review process. We sincerely appreciate your efforts and have carefully revised the manuscript in accordance with the comments received.

Reviewer #1 (Remarks to the Author):

Authors addressed all issues nicely and have substantially revised the manuscript. I do not have further criticisms on the revised version.

Response: We appreciate your positive assessment and thank you for your valuable comments throughout the revision process.

Reviewer #2 (Remarks to the Author):

The authors have addressed most of my comments. There are two outstanding critique points:

1a. The new metabolomics data is not publically available.

1b. It is unclear from the Methods what level of evidence was used for metabolite annotation (retention time? &/ MS1? &/ MS2?).

Response: We apologize for the earlier omission. The metabolite annotation was conducted based on MS2 spectra using the commercial in-house BiotreeDB v3.0 database, and this information has now been included in the revised Methods section (**Line 539**). Furthermore, we have deposited the raw metabolomics data in the MetaboLights database, as indicated in **lines 709–710**. Thank you very much for your reminder.

2. regarding my previous comment, "Fig 2C – These seem to me like ‘sanity check’ results rather than revealing new biological information, especially since the metabolic models were refined using the BiOLOG data.": The authors have misunderstood the point. The point is that if BioLOG data is used to build. models, it is a circular argument to use the same data to validate.

Response: Thank you for the clarification, and we now understand it. In the revised version, we have redrawn Fig. 2c & 2e using MIP and MRO values derived from metabolic models **not refined** by BIOLOG data, with results consistent with the original (**See lines 168-173, 489-490, 496-499, and revised Fig. 2**). In addition, we have retained the validation of the pattern whereby narrow-spectrum resource-utilizing strains enhance community interactions (Lines 182-224), as this supports the generalizability of the proposed construction strategy. We appreciate your patience and helpful guidance.

Reviewer #3 (Remarks to the Author):

Response: Thank you for your kind contribution and co-review of our manuscript.

Reviewer #2 (Remarks to the Author):

I am afraid that the authors have missed again the point on the use of BIOLOG data for model building/validation. There is absolutely nothing wrong in using BIOLOG to build the model. In fact, it is highly recommended and should be used. My point is that consistency of the resulting models with the data used to build the models should NOT be phrased as "validation". I suggest only to rephrase along these lines, not to revise figures using models that do not use BIOLOG data.

Response: Thank you for your patience and valuable suggestions. In the revised version, we have restored Fig. 2c & 2e to their original form, which is based on Biolog-refined models. Importantly, we have revised the main text (**Lines 145-178**) and the caption of Fig. 2 (**Lines 711-725**) to **avoid “validation”** according to your suggestions. If there is anything we may have overlooked or misinterpreted, we would greatly appreciate your further guidance. Thank you very much.

Reviewer #2 (Remarks to the Author):

The authors have addressed my remaining concern. Congratulations on a nice study.

Response: Thank you for your feedback and kind words. We appreciate your support for our study.